# Detection and Characterization of Visceral Anisakid Nematodes in Blue Whiting from Portuguese Waters

**DOI:** 10.3390/foods13233802

**Published:** 2024-11-26

**Authors:** Athanasia Rigkou, Mahima Hemnani, Ana Luísa Martins, João R. Mesquita

**Affiliations:** 1Department of Biomedical Sciences, School of Health and Care Sciences, University of West Attica, 12243 Athens, Greece; athanasiarigkou@gmail.com; 2School of Medicine and Biomedical Sciences, Porto University, 4050-313 Porto, Portugal; hemnanimahi@gmail.com (M.H.); analuisam11@gmail.com (A.L.M.); 3Epidemiology Research Unit (EPIUnit), Instituto de Saúde Pública da Universidade do Porto, 4050-091 Porto, Portugal; 4Laboratório para a Investigação Integrativa e Translacional em Saúde Populacional (ITR), 4050-600 Porto, Portugal

**Keywords:** *Anisakis simplex*, *Anisakis pegreffii*, *Hysterothylacium aduncum*, *Micromesistius poutassou*, seafood safety, public health

## Abstract

This study employs molecular detection techniques, including conventional PCR and Sanger sequencing, to investigate the prevalence, species composition and public health implications of Anisakid nematodes in blue whiting (*Micromesistius poutassou*) caught off the Portuguese coast. With Portugal’s high fish consumption rates and increasing preference for raw or undercooked seafood, the risk of parasitic infections, particularly anisakidosis, is rising. Fifty blue whiting fish were examined, showing a 100% infection rate with Anisakid larvae. Molecular analysis identified 68.1% of the larvae as *Anisakis simplex*, 18.1% as *Anisakis pegreffii*, and 13.8% as *Hysterothylacium aduncum*, marking the first report of *H. aduncum* in blue whiting in Portugal. Phylogenetic analysis based on the internal transcribed spacer (ITS) 1, 5.8S ribosomal RNA and ITS-2 confirmed the species classification. Notably, 42.9% of the fish were infected with multiple Anisakid species, increasing the risk of allergenic sensitization. Statistical analysis showed no significant correlation between fish width and parasitic load, and a weak negative correlation was found between fish length and parasitic load. The study contributes to food safety by integrating molecular tools that enable rapid and accurate species identification, offering new insights into the detection of biological contaminants in seafood. These findings are significant considering the rising trend in raw seafood consumption, underscoring the urgent need for enhanced detection strategies and broader parasite monitoring programs to mitigate public health risks. The high prevalence of parasitized fish highlights the necessity for the implementation of safe cooking practices to reduce the risk of anisakidosis. Further research into the allergenic potential of *Hysterothylacium* spp. and the ecological factors influencing this nematode distribution is recommended.

## 1. Introduction

Portugal is one of the countries with the highest rates of seafood consumption worldwide [1,2]. Traditionally, Portuguese cuisine emphasizes fully cooked or processed fish dishes due to a strong culinary culture and abundant marine resources [3]. However, recent trends influenced by globalization are introducing new eating habits, including an increased preference for undercooked and raw fish, like sushi and ceviche, into the Portuguese food landscape [4,5]. This shift in dietary habits has raised concerns about the risk of exposure to parasitic infections, given the major risk factor that foodborne parasites pose [6].

Parasitic nematodes of the families Anisakidae and Raphidascarididae, commonly referred to as Anisakids, are the main cause of anisakidosis, a fish-borne zoonotic disease that can infect humans [7,8]. Anisakidosis has started to pose a significant public health concern, particularly in Portugal, due to the increased consumption of raw or undercooked seafood, which can lead to the accidental ingestion of Anisakid larvae [1,5]. The introduction of these new fish consumption habits heightens the risk of exposure to parasites from the Anisakidae and Raphidascarididae families, which are among the most abundant parasites in marine fish worldwide [9]. This can lead to a range of symptoms, from acute or chronic gastrointestinal distress to severe allergic reactions [10].

Anisakidosis manifests in various clinical forms, with gastric and intestinal types being the most frequent. These forms arise when larvae infiltrate the stomach or intestinal lining after accidental ingestion, leading to sharp abdominal pain and inflammation and potentially the formation of granulomas due to the immune response to the larvae. In the rarer ectopic anisakidosis, larvae migrate to non-gastrointestinal tissues, such as the pharynx, lungs, or peritoneal cavity, where they cause localized inflammatory reactions [1]. Beyond gastrointestinal symptoms, anisakidosis can trigger severe allergic responses, known as gastroallergic anisakidosis [11,12]. 

The genus *Anisakis* includes at least nine distinct species, categorized into two complexes: the *Anisakis simplex* and the *Anisakis physeteris* complexes. The *A. simplex* complex includes *A. simplex* sensu stricto, *Anisakis pegreffii*, and *A. simplex* C, while the *A. physeteris* complex includes *A. physeteris*, *Anisakis typica*, *Anisakis paggiae*, *Anisakis ziphidarum*, *Anisakis brevispiculata*, and *Anisakis nascettii* [13,14]. The most prevalent *Anisakis* species causing human infections are *A. simplex* and *A. pegreffii* [12].

There have also been case reports from Spain detecting the genus *Hysterothylacium* of the family Raphidascarididae, which, while not as pathogenic, can also cause allergic reactions [15]. Nematodes from this genus are widely distributed parasites in diverse marine organisms and are the most abundant and diverse group of ascaridoids infecting marine fish globally, comprising around 72 species [16]. 

The life cycle of Anisakid nematodes involves several stages and multiple hosts [17,18,19]. Adult worms inhabit the stomach and intestine of marine mammals, where they reproduce and release eggs into the ocean via feces. These eggs hatch into stage two larvae (L2) on the ocean floor and are ingested by planktonic crustaceans, which serve as the first intermediate hosts, allowing the larvae to develop into stage three larvae (L3). Fish or cephalopods act as paratenic hosts by consuming these infected crustaceans, with the L3 larvae residing in the viscera and possibly migrating to the muscle without further development [8]. The paratenic hosts are crucial in transferring the larvae to their definitive hosts, cetaceans, where they molt into stage four larvae (L4) and mature into adult worms [20,21]. Humans become accidental hosts by consuming raw or undercooked infected fish or cephalopods, interrupting the life cycle since the larvae cannot reproduce in humans [22]. Similarly, the genus *Hysterothylacium* follows a comparable life cycle, with adult worms maturing in the stomach or intestine of fish, using crustaceans as the first intermediate hosts [23]. Larvae of *A. simplex* and A. pegreffii are predominantly found within the musculature, body cavity, and viscera of infected fish, whereas *Hysterothylacium aduncum* primarily resides in the viscera and visceral cavity [24]. 

In cetaceans, adult nematodes can cause significant health issues, including ulcers, severe infections, hemorrhage, stomach perforation, peritonitis, and even death [20]. In humans, the symptoms of anisakidosis can range from severe abdominal pain, which may require surgical intervention, to acute allergic reactions such as urticaria or anaphylaxis, often resulting in vomiting or diarrhea due to the larvae penetrating the stomach or intestine [13,25]. These diverse clinical and allergic responses underscore the public health importance of anisakidosis, especially in populations with high seafood consumption. 

Understanding the immune responses of each host to parasitic infections is crucial for elucidating the mechanisms that govern host resistance and susceptibility. *Anisakis* species, such as *A. simplex* and *A. pegreffii*, are notable for their complex interactions with fish hosts and, to some extent, humans that can act as incidental hosts. These parasites can alter the host’s immune response by secreting immunomodulatory molecules that interfere with immune signaling pathways, thereby preventing the host’s ability to mount an effective immune response and clear the parasite [26].

Similarly, *Hysterothylacium* spp., such as *H. aduncum*, interacts with the host’s immune system in distinct ways. While *H. aduncum* is generally viewed as less harmful to humans compared to the *Anisakis* species, its ability to trigger allergic reactions remains an area of ongoing research, highlighting the necessity for additional monitoring to fully understand parasitic diversity and potential health implications [27]. 

Blue whiting fish (*Micromesistius poutassou*) is widely consumed in the northeast Atlantic Ocean and the west Mediterranean Sea [28]. This fish primarily feeds on small crustaceans and cephalopods, which are paratenic hosts for Anisakidae larvae, leading to high parasitization rates [29]. Due to its wide availability and low cost, blue whiting fish consumption is prevalent in southwestern Europe [30]. 

Blue whiting is deeply embedded in the Portuguese diet due to its abundance in local waters, affordability, and versatility in traditional cuisine [31]. However, its high consumption rates raise concerns regarding food safety, particularly in relation to Anisakid nematodes, which are among the most prevalent parasites in marine fish globally [32,33]. Despite this fish species being one of the most consumed fish in Portugal, there has been a lack of comprehensive studies addressing the prevalence of these parasitic nematodes in them. 

Given the growing trend of raw seafood consumption and the frequent use of blue whiting in various dishes, testing this species for Anisakid nematodes is crucial to the marine ecosystem and, by extension, to the human population [34]. Anisakidosis is increasingly reported worldwide due to rising seafood consumption, particularly in Europe and Asia [34]. In Portugal, anisakidosis has become a public health concern, especially as blue whiting is commonly consumed both cooked and raw. Routine testing for Anisakid contamination in blue whiting not only safeguards public health by identifying potential infection risks but it ensures that the high demand for this affordable fish does not compromise consumer safety as well. Identifying and managing these biological contaminants through routine testing is essential to maintaining the safety and sustainability of this key food resource in Portugal [1].

In this context, studies like the current research are crucial for evaluating the prevalence of nematode infections in blue whiting fish. This study aims to accurately identify the species of Anisakid nematodes present in blue whiting fish in Portugal. This is essential as these nematode families include parasites with varying levels of pathogenicity, impacting both human and veterinary health while also adding to the understanding of parasite distribution in marine environments and its implications for food safety.

This study utilizes molecular techniques, such as conventional PCR coupled with Sanger sequencing, to provide accurate species identification, contributing to the development of rapid and reliable methods for detecting biological contaminants in seafood. 

## 2. Materials and Methods

### 2.1. Sample and Data Collection

Blue whiting fish (*M. poutassou*) (*n* = 50) caught off the Portuguese coast and intended for human consumption were purchased in the city of Porto, Portugal, in March 2024 and transported to the laboratory under refrigerated conditions. This species is particularly abundant in the northeast Atlantic, especially along the continental shelves of Portugal and surrounding regions, contributing to its accessibility and affordability in local markets [30]. Due to its widespread availability and relatively low cost, blue whiting is a staple in the Portuguese diet, and its high consumption underscores its relevance in assessing potential health risks. Thus, investigating the potential risk of anisakidosis linked to Anisakid species in blue whiting was a priority in this study, given the public health implications associated with consuming this commonly eaten fish [30]. 

Each fish was measured in length and width, carefully dissected, and the visceral organs were examined for the presence of Anisakid larvae. The larvae from the viscera were first counted and then extracted. A total of two larvae, the smallest and the largest of each fish, were collected and preserved at −20 °C for subsequent molecular analysis. Collecting the smallest and largest larvae from each fish likely increases the chances of capturing a broader spectrum of genetic diversity among the parasites, as size variation likely reflects distinct infections in time and hence differences in genetic lineages. 

### 2.2. Nucleic Acid Extraction

For DNA extraction, each sample was initially disrupted to ensure thorough homogenization using a mortar and pestle [35]. Additionally, we used a combination of mechanical disruption methods, including bead beating and vortexing. Each sample was mixed with sterile beads and 400 µL of Buffer ATL Tissue lysis buffer, then vortexed vigorously in a disruptor genie for 5 min to break down the larval tissue.

Following mechanical disruption, the samples were incubated at 57 °C for 24 h to ensure complete lysis of the tissue. Total nucleic acid was extracted from 700 µL of the resulting mixture, which was combined with RNase-free water. The extraction was carried out using the QIAamp Viral RNA Mini Kit (Qiagen, Hilden, Germany, reference number: 133226410), following the manufacturer’s protocol. This method ensured the efficient recovery of high-quality nucleic acids suitable for downstream applications.

### 2.3. Molecular Detection of Anisakid Larvae

For the molecular detection of Anisakid larvae, the conventional PCR kit from GRiSP^®^, Porto, Portugal, was employed, using the forward primer NC5 (5′-GTA GGT GAA CCT GCG GAA GGA TCA TT-3′) and the reverse primer NC2 (5′-TTA GTT TCT TTT CCT CCG CT-3′) which target the entire ITS region (ITS-1, internal transcribed spacer 1, 5.8S ribosomal RNA and ITS-2, internal transcribed spacer 2) [7,36]. Briefly, 1 μL of the extracted DNA product was utilized as a template using the Xpert Fast Hotstart Mastermix (2 x) with dye (GRiSP^®^, Porto, Portugal). A final amount of 25 μL was used for the PCR reaction mix. The following conditions were used in the Veriti 96-well thermal cycler (Thermo Fisher Scientific Inc., Waltham, MA USA) for amplification reactions with positive and negative controls: an initial cycle of 5 min at 95 °C, followed by 40 cycles of 94 °C for 2 s, 56 °C for 5 s, and 72 °C for 5 s, with a final elongation at 72 °C for 10 min.

The PCR products were run on a 1% agarose gel stained with Xpert Green Safe DNA gel stain (Grisp, Porto, Portugal) at 120 V for 30 min. The targeted DNA fragments were identified based on their molecular weights in comparison to a DNA ladder (Grisp, Porto, Portugal) and were visualized under UV light to confirm and validate the findings (Appendix A). 

### 2.4. Sanger Sequencing and Phylogenetic Analysis

The amplicons were purified using the Exo/SAP Go PCR Purification Kit (Grisp, Porto, Portugal), in accordance with the guidelines given by the manufacturer. Sanger sequencing was then performed using specific forward primers which target the entire ITS region [7]. The sequences were processed and aligned using a BioEdit Sequence Alignment Editor version 7.2 (Ibis Biosciences, Carlsbad, CA, USA) and compared to those in the NCBI nucleotide database (GenBank, Carlsbad, CA, USA) via the NCBI BLAST tool (https://blast.ncbi.nlm.nih.gov/Blast.cgi, accessed on 25 May 2024).

Sequences from this study were submitted to GenBank with accession numbers PP844718-PP844781, PP845310-PP845326, and PP845329-PP845341. These, along with reference strains from the Anisakidae (*A. simplex* and *A. pegreffii*) and Raphidascarididae family (*H. aduncum*) from GenBank, were aligned using MEGA X software (https://www.megasoftware.net/, accessed on 25 May 2024).

The optimal model, in MEGA software (https://www.megasoftware.net/, accessed on 25 May 2024), was chosen based on the lowest Bayesian Information Criterion (BIC) score, using the maximum likelihood method with the general time reversible model and a discrete Gamma distribution, assuming invariable sites with 1000 bootstrap replicates.

The phylogenetic trees were edited and visualized using the Interactive Tree of Life (iTOL) platform, (https://itol.embl.de/, accessed on 15 June 2024) providing detailed graphical representation and annotation of the phylogenetic relations among the analyzed sequences.

### 2.5. Statistical Analysis

To analyze the correlation between fish morphometrics (length and width) and parasitic load, we conducted a Pearson correlation test using the R computational environment (version 10) [37]. Before performing the Pearson correlation test, we used the Shapiro–Wilk test to check for normality of the data. We selected the Pearson correlation test because it measures the linear relationship between continuous variables, which is appropriate for our data that exhibits linear relationships and follows a normal distribution. The ggplot2 R package was used for visualization.

## 3. Results

From the total 50 *M. poutassou,* all were shown to be parasitized with Anisakid larvae. The average parasitic load was 29.20, with a maximum of 400 and a minimum of 3. The average length of the fish was 21.31 cm, ranging from 16 to 23.5 cm. The average width was 3.6 cm, with a range from 2 to 5.5 cm. From the total 100 Anisakid samples selected and tested, 94 produced sequences with sufficient quality to determine their species. Sequencing and subsequent nucleotide BLAST analysis indicated that 68.1% (*n* = 64) of the samples were identified as *A. simplex*, 18.1% (*n* = 17) as *A. pegreffii*, and 13.8% (*n* = 13) as *H. aduncum*. We observed 21 fishes (42.9%) infected with more than one species of Anisakid larvae (Appendix A).

The retrieved sequences were assigned the following accession numbers: PP844718-PP844781 (*A. simplex*), PP845310-P845326 (*A. pegreffii*), PP845320-PP845341 (*H. aduncum*). Additional characterization through BLAST analysis revealed that the obtained sequences for *A. simplex* clustered with reference sequences from various hosts and locations. Specifically, the sequences showed 100% identity with samples from Greenland obtained from the fish *Gadus morhua*, 98.16–98.22% identity with samples from Portugal obtained from the fish *Aphanopus carbo*, 99.70% identity with a sample from the Aegean Sea coast of Turkey obtained from the fish *Zeus faber*, 100% identity with samples from Spain, 99.85% identity with a sample from the fish *Scomber scombrus*, and 97.96–100% identity with samples retrieved from a human. The sequences obtained for *A. pegreffii* clustered with reference sequences from Japan, also retrieved from a human, showing percentage identities ranging from 99.00% to 100%. For *H. aduncum*, the sequences clustered with reference samples obtained from the fish *Engraulis encrasicolus* found in the Mediterranean Sea and Adriatic Sea, showing identities ranging from 99.81% to 100%.

Phylogenetic analysis using the obtained sequences and 11 reference strains confirmed the classification for each one of the species, as illustrated in Figure 1.

We also analyzed the correlation between fish morphometrics and parasitic load. Pearson’s correlation coefficient was calculated to assess the relationship between the width/length of the fish and their parasitic load. For the width and the parasitic load, the test showed a correlation of 0.0878 (IC = −0.1239 to 0.2919; *p*-value = 0.416). This shows that the correlation is not statistically significant (Figure 2) as the *p*-value of 0.416 is more than the threshold of 0.05, implying no significance.

Analyzing the relationship between fish length and parasitic load, we found a significant, but weak correlation (IC = −0.4072 to −0.0070; *p*-value = 0.0431). The *p*-value is less than the threshold of 0.05, which implies that as the length of the fish increases, the parasitic load tends to decrease (Figure 3).

## 4. Discussion

Anisakidosis, also known as anisakiosis, is a fish-borne zoonotic disease that can pose a risk to human health, when raw or undercooked fish infested with Anisakid larvae are consumed [8]. When ingested, the larvae can penetrate the gastrointestinal tract, leading to various symptoms ranging from mild to severe [10]. 

Due to the widespread consumption of raw or undercooked seafood, which is increasingly popular in many cuisines worldwide, anisakidosis constitutes a significant global health hazard. In countries with high seafood consumption, thousands of cases are reported each year, often linked to cultural and culinary practices that favor raw or lightly cooked fish [38]. Recent data suggests that over 20,000 cases of anisakidosis are reported annually worldwide, with the majority occurring in regions such as Japan, Spain, and parts of Northern Europe [5] However, the demand for dishes with minimally processed seafood has grown internationally, expanding anisakidosis risk to countries where infection rates were previously low, such as Portugal, as consumer preferences adapt to globalized culinary trends [1]. Addressing anisakidosis as a global threat underscores the need for further studies to understand and mitigate its impact on public health. 

Anisakid larvae, in addition to gastrointestinal symptoms, release allergens during their life cycle that can provoke immune responses, leading to allergic symptoms, which can occur either through ingestion or even direct handling of infected fish. Exposure to these nematode species may increase allergy risks, and repeated contact with allergenic proteins can sensitize individuals over time [39]. This aspect of anisakidosis adds a layer of complexity to the health risks associated with Anisakid infections, further underscoring the need for awareness and preventive measures among consumers and food safety authorities. 

Besides the health concerns that anisakidosis poses to individual consumers, it also has broader implications for the fishing industry and seafood consumption patterns. In regions where seafood is a dietary staple, such as Portugal, Japan, and parts of Spain, even low public awareness of anisakidosis can significantly influence consumer behavior and market demand [39]. A higher rate of anisakidosis cases has the potential to decrease consumer confidence in seafood safety, leading to reduced fish sales, particularly of species known to carry Anisakid larvae, such as blue whiting and other small pelagic fish [32,38]. Consequently, balanced approaches from public health authorities that prioritize both consumer safety and industry sustainability are required. 

Our research primarily focused on the presence of Anisakid larvae in the viscera of blue whiting fish (*M. poutassou*) in Portuguese waters. Although the larvae are typically concentrated in the intestines, it is important to note that larvae may migrate into the muscle tissue, which is the edible part of the fish, thus creating potential implications for food safety and consumer health [40]. This could increase the risk to consumers, as the muscle tissue is commonly consumed and may harbor viable larvae or their allergens. While the larvae found in the viscera alone do not directly pose a health risk if the organs are removed and properly handled during processing, there is growing concern that improper handling or consumption of undercooked fish may result in the ingestion of larvae from the muscle tissue [41]. 

Our study provides new insights into the prevalence and distribution of Anisakid nematode species in blue whiting fish caught off the coast of Portugal. The application of molecular techniques such as conventional PCR and Sanger sequencing in this study provides a rapid and accurate approach for identifying biological contaminants, which is critical for preventing zoonotic diseases like anisakidosis and safeguarding public health. 

In total, 50 blue whiting fish were examined, showing a 100% infection rate with Anisakid larvae. Molecular analysis identified 68.1% of the larvae as *A. simplex*, 18.1% as *A. pegreffii*, and 13.8% as *H. aduncum*, marking the first report of *H. aduncum* in blue whiting in Portugal. The 100% infection rates observed are consistent with previous reports on Anisakid nematode infections in marine fish [42,43]. *Anisakis simplex* was the most prevalent species identified in blue whiting, in agreement with findings from similar studies that report this parasite species as the predominant nematode species in various marine fish hosts, establishing it as the primary etiological agent responsible for anisakidosis [44,45].

Consuming blue whiting infected with multiple species of Anisakid nematodes may increase the exposure to a broader spectrum of allergens. Certain allergens from Anisakids, including both native and recombinant forms of tropomyosin and paramyosin, have shown significant cross-reactivity with similar proteins found in other invertebrates, such as crustaceans and mites, hence increasing the risk of allergic reactions in humans [46].

Two larvae, the largest and the smallest, were collected from each fish to increase the chances of capturing a broader spectrum of genetic diversity among the parasites. By collecting larvae of varying sizes, researchers can gain a more comprehensive understanding of the diversity of parasitic species present and can account for the temporal dynamics of infection, as smaller larvae may represent more recent infections, while larger larvae could be indicative of earlier infections [47]. This approach allows for a more accurate assessment of associated risks, such as allergenic sensitization, linked to the consumption of infected fish. Following this methodology, we observed that 42.9% of the blue whiting (*n* = 21) were infected with more than one species of Anisakids, which could highlight the higher likelihood for allergenic sensibilization to human health, when consuming this fish species. It is noteworthy that the width of blue whiting fish did not have a statistically significant impact on the parasitic load of Anisakid nematodes. In contrast, a negative correlation was detected between the length of the fish and the parasitic load, albeit weak. These results show that no conclusion can be taken regarding the correlation between the width of the fish and the parasitic load. However, the inverse correlation between fish length and parasitic load suggests that larger fish tend to harbor fewer parasites. This finding contrasts with previous studies that reported positive correlations regarding fish length and parasitic load [7,48], implying that factors other than size may play a crucial role in determining parasitic load. Further research is needed to elucidate the mechanisms behind these correlations and to understand the ecological and biological factors influencing parasitic infections in blue whiting. 

Portugal stands among the world’s highest consumers of seafood, with fish playing a central role in its culinary culture [5]. Traditionally, Portuguese diets have emphasized fully cooked or processed fish, but recent global influences have introduced a preference for raw or lightly processed fish dishes [3]. This change mirrors a broader global shift towards minimally processed seafood, reflecting evolving dietary preferences across many cultures.

Although the number of confirmed anisakidosis cases in Portugal remains relatively low and has not raised significant public health concerns, there are indications that the actual risk to consumers may be underestimated [5]. Anisakidosis is often challenging to diagnose due to its wide range of non-specific symptoms, which can overlap with other gastrointestinal conditions, leading to frequent misdiagnosis or underreporting [49]. This suggests that the true infection rate of Anisakid nematodes could be higher than current statistics indicate, underscoring the need for increased diagnostic awareness and public health surveillance. 

Our study observed a high rate of Anisakid parasitism in blue whiting, which aligns with reports of heavy parasitic loads in other widely consumed species in Portugal, such as anchovies, tuna, sardines, mackerel, and hake [2,33,50]. This similarity underscores a broader trend of high anisakid prevalence across economically important fish in the northeast Atlantic, suggesting that *Anisakis* spp. infection may be pervasive in the region’s marine ecosystem. However, blue whiting remains relatively underexplored as a vector for anisakidosis in this region. 

This comparative approach reinforces the significance of our findings by shedding light on the anisakid infection rates in blue whiting, a lesser-studied yet widely consumed species in Portugal. Detecting nematode larvae in blue whiting, especially within the visceral organs, is crucial given its high consumption and, by extension, its relevance to public health. *Anisakis simplex* was the most detected Anisakid species in our study, emphasizing the critical need for proper cooking methods to mitigate the risk of infection in humans. Ensuring proper culinary practices is crucial to safeguard public health, especially in species like blue whiting, which are intended for human consumption.

The impact of *Anisakis* spp. on human health, particularly in relation to allergic reactions, has drawn increasing attention in recent years. Allergic reactions to *Anisakis* spp. can occur following the consumption of both live and dead larvae, provoking symptoms such as urticaria, angioedema, or anaphylaxis. These symptoms are mediated by immunoglobulin E (IgE), which plays a central role in type I hypersensitivity reactions as well as immune responses to parasitic infections. Notably, even after the larvae die, allergenic proteins persist in the fish tissues, meaning that properly cooked or frozen fish can still trigger severe allergic reactions in sensitized individuals [50].Cross-reactivity with proteins from other nematodes, or between different nematode species, further complicates diagnosis, as individuals may exhibit positive reactions to Anisakid antigens without true sensitization [46]. 

Additionally, *Anisakis* spp. provoke a Th2 immune response in all individuals infected, regardless of their allergic predisposition, indicating that their influence on the immune system may go beyond typical allergic responses. Despite advances in understanding the allergens produced by these parasites, the nature of their interaction with humans, who are accidental hosts, is not fully understood. Thus, further investigation into non-allergic *Anisakis* infections and their potential long-term immunological effects is essential for advancing diagnostic tools and improving public health measures [51].

This study is particularly significant as it marks the first recorded occurrence of *H. aduncum* in blue whiting fish (*M. poutassou*) from Portuguese waters, providing new insights into the geographic and host distribution of this parasite. The findings of *H. aduncum* in this fish species within this specific region not only broaden the understanding of parasitic presence in marine ecosystems but also underscore the potential implications for food safety, especially given blue whiting’s importance in the Portuguese diet. Identifying this parasitic species in a heavily consumed fish like blue whiting emphasizes the need for heightened surveillance and risk management to protect public health. 

The discovery of *H. aduncum*, although not commonly associated with human infections, highlights the necessity of monitoring for a wider range of parasites [23]. Although *Hysterothylacium* spp. are generally considered less pathogenic to humans compared to *Anisakis* spp., their potential to provoke allergic reactions remains a topic of interest. Previous studies have reported cross-reactivity of *Anisakis* allergens with homologous proteins in other invertebrates, suggesting that similar mechanisms might exist for *Hysterothylacium* spp. [27]. This raises the possibility of a lower allergenic potential, though further research is needed to confirm this.

The detection of *H. aduncum* in blue whiting fish off the coast of Portugal suggests a broader distribution and potential adaptation of this parasite in different marine environments. Environmental factors such as habitat structure, species biodiversity, and oceanographic conditions likely play a pivotal role in shaping the prevalence and distribution of these nematodes. Understanding these ecological factors is crucial for effectively managing the consequences of parasite transmission on both marine ecosystems and public health. 

This study enhances our understanding of the prevalence, distribution, and potential health risks associated with nematode parasites in blue whiting fish from the coast of Portugal. These findings emphasize the need for continuous monitoring and improved cooking practices to safeguard public health. Further research into the allergenic potential of *Hysterothylacium* spp. and the ecological factors influencing nematode distribution is crucial for fully grasping their impact and devising effective mitigation strategies. 

The present research addresses a critical gap in seafood safety studies by focusing on Anisakid nematode infection risks in blue whiting (*M. poutassou*), a species widely consumed in Portugal but often overlooked in parasite risk assessments. Unlike previous studies that broadly examine parasitic nematode presence in various fish species, this research specifically investigates blue whiting, accounting for its important role in Portuguese diets. By applying advanced diagnostic techniques and analyzing samples from the Portuguese coast, our findings provide novel insights into species-specific risks of anisakidosis, with direct implications for public health policy and food safety practices.

Future studies on the immune responses of blue whiting to Anisakid nematode infections are also crucial for understanding host–parasite interactions and the factors that modulate infection outcomes. Gaining insights into these immune mechanisms can facilitate the development of strategies to reduce parasitic burdens, thereby improving fish health and minimizing the risks to human consumers. Such knowledge could contribute to enhancing disease resilience in wild fish stocks and aquaculture populations, ultimately supporting the health and sustainability of marine ecosystems within the food chain.

## 5. Conclusions

This study revealed the widespread prevalence of Anisakid nematodes in blue whiting fish (*M. poutassou*) caught off the Portuguese coast, with all examined fish found to be parasitized. The molecular identification of *A. simplex*, *A. pegreffii*, and the first report of *H. aduncum* in blue whiting from Portugal underscores the importance of ongoing monitoring, particularly given the rising popularity of undercooked or raw fish consumption. The detection of *H. aduncum* in blue whiting represents the first such report in Portugal, expanding our understanding of its geographic distribution. While no significant correlation was found between fish width and parasitic load, a weak negative correlation between fish length and parasitic burden indicates that smaller fish may be more heavily infested with parasites. Given the high rates of parasitic infection of this fish species in Portuguese waters, our findings underscore the need for enhanced surveillance and regulatory measures to mitigate the risks of zoonotic diseases and safeguard public health.

These findings highlight the need for enhanced food safety practices, particularly in ensuring thorough cooking to reduce the risk of anisakidosis. Furthermore, the detection of multiple Anisakid species in nearly half of the sampled fish suggests the potential for heightened allergenic sensitization to human consumers, warranting further research into the ecological and public health implications of such parasitic infections.

The application of molecular techniques, such as PCR and Sanger sequencing, represents a novel and rapid detection method for biological contaminants in seafood, offering a reliable tool for improving food monitoring and safety practices.

## Figures and Tables

**Figure 1 foods-13-03802-f001:**
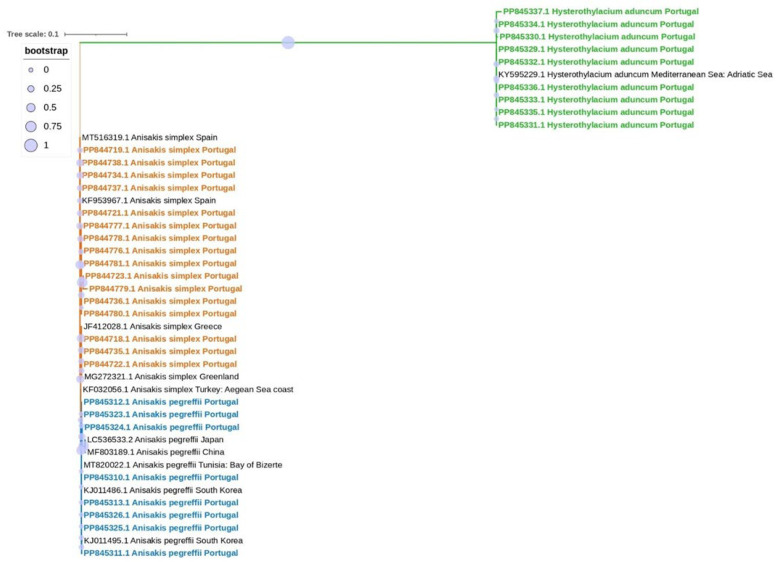
Phylogenetic tree constructed for *Anisakis simplex*, *Anisakis pegreffii*, and *Hysterothylacium aduncum* using 11 global reference strains. Phylogenetic analysis was based on the entire 950 bp ITS region. The tree was constructed using MEGA X software, employing the maximum likelihood method based on the Jukes–Cantor model. The robustness of the tree was assessed with 1000 bootstrap replicates. Samples from this study are indicated in bold. Samples indicated in orange represent *A. simplex*, in blue represent *A. pegreffii* and in green represent *H. aduncum*. Each sample is labeled with its species, GenBank accession number, and country of origin.

**Figure 2 foods-13-03802-f002:**
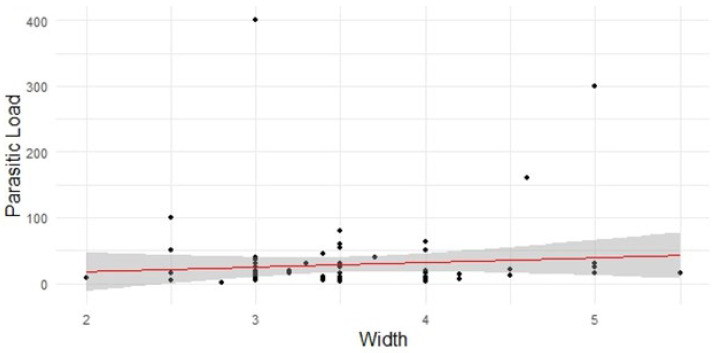
Scatter plot illustrating the relationship between fish width and parasitic load. The shaded region represents the confidence interval around the regression line.

**Figure 3 foods-13-03802-f003:**
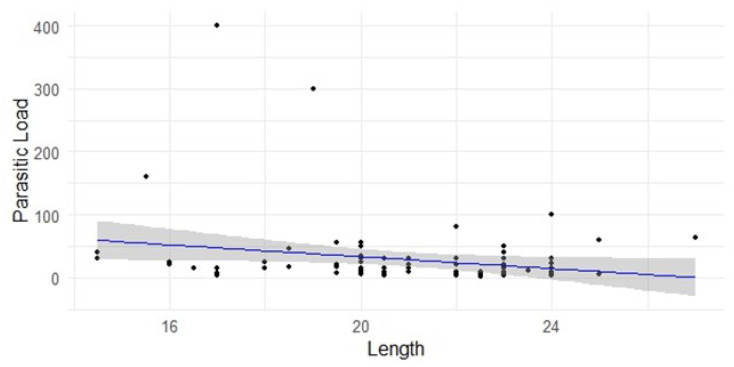
Scatter plot illustrating the relationship between fish length and parasitic load. The blue regression line indicates a slight negative correlation, suggesting a trend where parasitic load decreases as fish length increases. The shaded region represents the confidence interval around the regression line.

## Data Availability

The data presented in this study are available on request from the corresponding author.

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
