# Peer review of "Detection and Characterization of Visceral Anisakid Nematodes in Blue Whiting from Portuguese Waters"

_foods, 2024, doi:10.3390/foods13233802_

Round 1
Reviewer 1 Report
Comments and Suggestions for Authors
According to my understending, the article with this title is not suitable for publication.
Firstly, the title is misleading to the reader as it suggests a focus on food safety, while the article only examines fish viscera rather than the edible muscle tissue that people actually consume. The authors have done good work; however, they need to adjust their approach to emphasize the potential risk posed by the presence of a high number of nematode larvae in the fish intestines, as this could potentially translate to their presence in the muscle tissue. Highlighting this connection would strengthen the study’s relevance to food safety concerns. The authors conducted fascinating research on the viscera but failed to consider that larvae in the viscera do not pose a threat to consumers.
Ultimately, this article, under its current title, is not suitable for a journal like FOODS.
Author Response
Comment 1 :
According to my understanding, the article with this title is not suitable for publication.
Firstly, the title is misleading to the reader as it suggests a focus on food safety, while the article only examines fish viscera rather than the edible muscle tissue that people actually consume. The authors have done good work; however, they need to adjust their approach to emphasize the potential risk posed by the presence of a high number of nematode larvae in the fish intestines, as this could potentially translate to their presence in the muscle tissue. Highlighting this connection would strengthen the study’s relevance to food safety concerns. The authors conducted fascinating research on the viscera but failed to consider that larvae in the viscera do not pose a threat to consumers.
Ultimately, this article, under its current title, is not suitable for a journal like FOODS.
Response 1 :
Thank you so much for the insightful feedback regarding the title and the study’s relevance to food safety. We have modified the title to: Detection and Characterization of Visceral Anisakid Nematodes in Blue Whiting from Portuguese Waters, explicitly incorporating “visceral” to better reflect the study’s emphasis.
Additionally, we have expanded the discussion to address the significance of examining nematode presence in the viscera, even though it is not consumed directly. Specifically, we highlighted that the high concentration of larvae in the viscera may indicate a risk of migration to muscle tissue, which would indeed pose a food safety concern. This addition emphasizes the broader implications of visceral nematode prevalence in understanding potential risks to consumers.
We hope these changes make the manuscript's relevance to food safety clearer, as suggested. Thank you again for helping us strengthen the manuscript.
Reviewer 2 Report
Comments and Suggestions for Authors
The article described in this review investigates the prevalence and species composition of Anisakid nematodes in blue whiting (Micromesistius poutassou) caught off the Portuguese coast, employing advanced molecular detection techniques such as conventional PCR and Sanger sequencing. Given Portugal's high rates of fish consumption and a growing trend towards raw or undercooked seafood, the study addresses the escalating risk of parasitic infections, particularly anisakidosis. The results reveal a concerning 100% infection rate among the examined fish, with molecular analysis identifying Anisakis simplex, Anisakis pegreffii, and Hysterothylacium aduncum, the latter representing a first report in this species in Portugal. The study underscores the urgent need for enhanced detection strategies and broader parasite monitoring programs to safeguard public health, particularly in light of the high prevalence of parasitized fish and the associated risks of allergenic sensitization.
The article is well-written and importantly draws attention to a significant yet still under-recognized issue like anisakiosis. However, the authors have overlooked a few errors. Below are the points that the authors should address or consider for improvement:
1. Line 11: Please expand on this section a bit and provide information on the prevalence of the disease, its symptoms.
2. Line 133: Consider specifying "in March 2024" instead of "March 2024" for clarity.
3. Line 136: You might clarify what "high consumption rate" means—do you have specific data or sources to back this claim?
4. Line 139: The phrase "preserved in -20°C" should be reworded to "preserved at -20°C" for correct phrasing.
5. Line 153: You could clarify the temperature range; use "incubated at 57°C for 24 hours" rather than "at 57 °C for 24 hours."
6. Line 179: Instead of "the BLAST tool," you could say "the NCBI BLAST tool" for clarity.
7. Line 200-252: It is worth explaining what the result "0.0431" exactly means in the context of the analysis so that the reader has a fuller understanding of the significance of this value.
8. Line 250-349: In my opinion, the discussion requires significant improvements. It would be beneficial to start the discussion by emphasizing the general health implications of anisakidosis in a global context. The authors may introduce statistics regarding the number of cases or their increase in recent years, which could justify the significance of your research. It is also important to mention the implications not only for human health but also for the fishing industry and fish consumption. Instead of referring to previous studies only in the context of your results, the authors should consider a brief analysis that compares your findings with other studies on anisakids in marine fish. This could strengthen argument regarding the uniqueness or typicality of the obtained results. Further development of the topic of anisakid allergies could also be beneficial. You could describe how exactly the mechanisms of allergic reactions work and why contact with different species of anisakids may increase the risk of allergies. This will enable readers to better understand the issue.
Author Response
Comment 1 :
The article described in this review investigates the prevalence and species composition of Anisakid nematodes in blue whiting (Micromesistius poutassou) caught off the Portuguese coast, employing advanced molecular detection techniques such as conventional PCR and Sanger sequencing. Given Portugal's high rates of fish consumption and a growing trend towards raw or undercooked seafood, the study addresses the escalating risk of parasitic infections, particularly anisakidosis. The results reveal a concerning 100% infection rate among the examined fish, with molecular analysis identifying Anisakis simplex, Anisakis pegreffii, and Hysterothylacium aduncum, the latter representing a first report in this species in Portugal. The study underscores the urgent need for enhanced detection strategies and broader parasite monitoring programs to safeguard public health, particularly in light of the high prevalence of parasitized fish and the associated risks of allergenic sensitization.
The article is well-written and importantly draws attention to a significant yet still under-recognized issue like anisakiosis.
Response 1 :
Thank you so much for the encouraging assessment of our manuscript and your valuable feedback.
However, the authors have overlooked a few errors. Below are the points that the authors should address or consider for improvement:
Comment 2 :
Line 11: Please expand on this section a bit and provide information on the prevalence of the disease, its symptoms.
Response 2 :
We have expanded this section in the introduction, to include additional information on anisakidosis prevalence and symptoms.
Comment 3 :
Line 133: Consider specifying "in March 2024" instead of "March 2024" for clarity.
Response 3 :
Changed accordingly.
Comment 4 :
Line 136: You might clarify what "high consumption rate" means—do you have specific data or sources to back this claim?
Response 4 :
We have clarified what is by providing more information and references that support this claim.
Comment 5 :
Line 139: The phrase "preserved in -20°C" should be reworded to "preserved at -20°C" for correct phrasing.
Response 5 :
Reworded accordingly.
Comment 6 :
Line 153: You could clarify the temperature range; use "incubated at 57°C for 24 hours" rather than "at 57 °C for 24 hours."
Response 6 :
We have rephrased this line.
Comment 7:
Line 179: Instead of "the BLAST tool," you could say "the NCBI BLAST tool" for clarity.
Response 7 :
We have modified it accordingly.
Comment 8 :
Line 200-252: It is worth explaining what the result "0.0431" exactly means in the context of the analysis so that the reader has a fuller understanding of the significance of this value.
Response 8 :
We have added an explanation to clarify the significance of the result "0.0431" within the context of the analysis. Specifically, we note that a p-value of 0.0431 is less than the threshold of 0.05, indicating that the result is statistically significant. This suggests there is sufficient evidence to reject the null hypothesis, thereby supporting the relevance of the finding within the context of our analysis.
Comment 9:
Line 250-349: In my opinion, the discussion requires significant improvements. It would be beneficial to start the discussion by emphasizing the general health implications of anisakidosis in a global context. The authors may introduce statistics regarding the number of cases or their increase in recent years, which could justify the significance of your research. It is also important to mention the implications not only for human health but also for the fishing industry and fish consumption. Instead of referring to previous studies only in the context of your results, the authors should consider a brief analysis that compares your findings with other studies on anisakids in marine fish. This could strengthen argument regarding the uniqueness or typicality of the obtained results. Further development of the topic of anisakid allergies could also be beneficial. You could describe how exactly the mechanisms of allergic reactions work and why contact with different species of anisakids may increase the risk of allergies. This will enable readers to better understand the issue.
Response 9 :
Thank you for your insightful and constructive comments on the discussion section. We greatly appreciate your suggestions for expanding and strengthening this part of the manuscript.
As recommended, we have restructured the beginning of the discussion to emphasize the general health implications of anisakidosis on a global scale. We have included statistics regarding the increasing number of reported cases of anisakidosis worldwide, particularly in regions with high fish consumption that follow the growing trend of consuming raw or undercooked seafood. This provides a broader context for the significance of our findings and justifies the importance of our study to public health.
We have also expanded the discussion to address the implications of anisakidosis not only for human health but also for the fishing industry. We described how the presence of Anisakid nematodes in marine fish can have broader economic and social implications, particularly in regions with high seafood consumption. We explained how increased cases of anisakidosis can influence consumer confidence in seafood safety, potentially leading to reduced demand for fish species known to carry Anisakid larvae, such as blue whiting. This underscores the importance of balanced approaches from public health authorities, which should consider both consumer safety and the sustainability of the fishing industry.
We have revised the manuscript to include a brief analysis that compares our findings with other studies on Anisakid nematodes in marine fish. We highlighted the high rate of Anisakid parasitism in blue whiting, which aligns with similar findings in other widely consumed fish species in Portugal, such as anchovies, tuna, sardines, mackerel, and hake. This comparison helps place our findings within a broader context, suggesting that Anisakid infections are pervasive across economically important fish species in the Northeast Atlantic. By framing our results within this broader trend, we underscored the significance of our findings and the relevance of blue whiting as a species of concern in terms of public health.
We have provided a more detailed explanation of the mechanisms underlying allergic reactions to Anisakid larvae. This includes a description of the immunological processes involved in allergic reactions to Anisakis spp., which can lead to allergic symptoms such as gastrointestinal distress, urticaria, or anaphylaxis. We also discussed the potential for sensitization due to simultaneous exposure to different species of Anisakid nematodes, explaining how this may increase the risk of developing allergies, especially among individuals who frequently consume raw or undercooked fish. This added detail enhances the understanding of the public health implications of parasitic infections in marine fish.
We hope these additions and revisions help to better contextualize our findings and strengthen the manuscript. Thank you again for your valuable feedback, which has contributed significantly to improving the quality and relevance of the discussion.
We trust that the revisions made have clarified the manuscript's importance to food safety, and we appreciate your assistance in improving the overall quality of our work.
Reviewer 3 Report
Comments and Suggestions for Authors
The manuscript titled “Application of Molecular Detection Methods for Anisakid 2 Nematodes in Blue Whiting: Prevalence, Species 3 Identification, and Public Health Implications for Food Safety” is informative for local protection against foodborne parasitic infections. The discovery of new infective species by sequencing is exciting. It is recommended that journals consider publication if authors address the following issues.
1. Lines 65-81 of the introductory part of the manuscript are proposed to be merged into one paragraph.
2. The description of fish consumption trends and blue whiting consumption in Portugal in the introduction section lacks data support and literature citations.
3. Line 193, in addition to the version, the author is required to include information such as the developer of the R software.
4. Line 202, What are the numerical units of parasitic load?
5. Whether it's font size, figure clarity, or compactness, authors need to improve the quality of the figures in the manuscript.
6. Manuscript needs to maintain consistency in formatting, for example, why is the period in line 253 red? The authors need to double-check.
7. The sample size of the study was small and only 50 fish may not be representative of infection in the entire blue whiting stock.
8. The manuscript is slightly vague about the source of the 50 Blue Whiting fish samples, and it is recommended that the authors provide a detailed description or add a map of the sample distribution, which would make the study more referential.
9. The manuscript mentions gel electrophoresis testing, but the results of the electrophoresis have not been seen and may be considered to be provided as supplementary material.
10. Authors must describe the novelty of this research.
11. References are not formatted consistently, e.g., references 23, 29, and 31 do not have DOI.
Author Response
Comment 1 :
The manuscript titled “Application of Molecular Detection Methods for Anisakid 2 Nematodes in Blue Whiting: Prevalence, Species 3 Identification, and Public Health Implications for Food Safety” is informative for local protection against foodborne parasitic infections. The discovery of new infective species by sequencing is exciting. It is recommended that journals consider publication if authors address the following issues.
Response 1 :
We appreciate your positive feedback on the study’s significance and are grateful for the detailed recommendations.
Comment 2 :
Lines 65-81 of the introductory part of the manuscript are proposed to be merged into one paragraph.
Response 2 :
We have merged the paragraphs into a single one as suggested.
Comment 3 :
The description of fish consumption trends and blue whiting consumption in Portugal in the introduction section lacks data support and literature citations.
Response 3 :
We have added data and literature citations to support the description of fish consumption trends, particularly blue whiting, in Portugal. The introduction, as well as materials and methods, now include more detailed information on regional consumption patterns and the significance of blue whiting as a food source, backed by relevant studies, and we have added more citation to strengthen the manuscript and provide more comprehensive background information.
Comment 4 :
Line 193, in addition to the version, the author is required to include information such as the developer of the R software.
Response 4 :
We have updated and included the necessary information, as well as the reference for the R software.
Comment 5 :
Line 202, What are the numerical units of parasitic load?
Response 5 :
We have added the numerical units to the supplementary file.
Comment 6 :
Whether it's font size, figure clarity, or compactness, authors need to improve the quality of the figures in the manuscript.
Response 6 :
We appreciate your comment regarding figure quality. We have reviewed and improved the clarity and presentation of all figures in the manuscript.
Comment 7 :
Manuscript needs to maintain consistency in formatting, for example, why is the period in line 253 red? The authors need to double-check.
Response 7 :
Thank you for noting this. We have carefully reviewed the entire manuscript for consistency in formatting and corrected the highlighted issue on line 253, along with any other discrepancies.
Comment 8 :
The sample size of the study was small and only 50 fish may not be representative of infection in the entire blue whiting stock.
Response 8 :
Thank you for raising this important point regarding the sample size. We acknowledge that the sample size of 50 fish is relatively small, and as such, it may not fully represent the infection rates in the entire blue whiting population. However, given that blue whiting has not been extensively studied in Portugal, this research can be considered a preliminary investigation into the prevalence of Anisakid nematodes in this species and this particular region.
The relatively small sample size is a limitation, but it provides valuable initial insights that may serve as a foundation for future studies with larger sample sizes. The findings presented in this study should therefore be interpreted with caution, and we emphasize the need for further research to confirm these results and expand the scope of our understanding of Anisakid nematode infection in blue whiting in Portugal.
Comment 9 :
The manuscript is slightly vague about the source of the 50 Blue Whiting fish samples, and it is recommended that the authors provide a detailed description or add a map of the sample distribution, which would make the study more referential.
Response 9 :
Thank you for your valuable suggestion. The 50 blue whiting fish samples used in this study were purchased from a well-known supermarket chain in Porto, Portugal, and were intended for human consumption. To protect the confidentiality of the retailer and avoid any potential issues with disclosure, we have not explicitly mentioned the name of the supermarket in the manuscript.
Regarding the suggestion to include a map, since the fish samples were collected only from Porto, we believe that creating a map is not necessary in this case, as the sample collection is geographically limited. However, we have clarified the origin of the samples in the manuscript to ensure transparency regarding their source.
Comment 10 :
The manuscript mentions gel electrophoresis testing, but the results of the electrophoresis have not been seen and may be considered to be provided as supplementary material.
Response 10 :
We have now included the results of the gel electrophoresis testing as supplementary material, as you recommended. The results are provided in the supplementary section, ensuring that they are accessible without detracting from the main manuscript’s flow.
Comment 11 :
Authors must describe the novelty of this research.
Response 11 :
Thank you for your valuable feedback regarding the novelty of the research. We have now clarified the innovative aspects of our study in the discussion section.
This research fills a critical gap in seafood safety studies by focusing on the Anisakid nematode infection risks in blue whiting fish (M. poutassou), a species that is widely consumed in Portugal but has been often overlooked in previous parasite risk assessments in this region. While many studies examine parasitic nematode presence in various fish species, our study uniquely investigates the species-specific risks associated with blue whiting, which plays an important role in the Portuguese diet.
By employing advanced molecular diagnostic techniques and analyzing samples from the Portuguese coast, our research provides new insights into the specific risks posed by anisakidosis in blue whiting, with significant implications for public health policies and food safety practices in the region. We believe this focus on a lesser-studied fish species and its direct connection to food safety in Portugal highlights the novelty and relevance of our work.
Comment 12 :
References are not formatted consistently, e.g., references 23, 29, and 31 do not have DOI.
Response 12 :
We have reviewed the reference list and ensured consistent formatting throughout. We have corrected the formatting issues with references 23, 29, and 31, and added DOI information where available to comply with citation guidelines. Thank you once again for your constructive suggestions, which have significantly contributed to improving our work.
Round 2
Reviewer 3 Report
Comments and Suggestions for Authors
The manuscript has been significantly improved. I recommend to accept it in present form.